# Mobile Mammography Services and Underserved Women

**DOI:** 10.3390/diagnostics12040902

**Published:** 2022-04-05

**Authors:** Usha Trivedi, Toma S. Omofoye, Cindy Marquez, Callie R. Sullivan, Diane M. Benson, Gary J. Whitman

**Affiliations:** 1Rutgers New Jersey Medical School, 187 S W Orange, Newark, NJ 07103, USA; utrivedi197@gmail.com; 2Department of Breast Imaging, The University of Texas MD Anderson Cancer Center, 1515 Holcombe Blvd, Unit 1350, Houston, TX 77030, USA; tsomofoye@mdanderson.org (T.S.O.); cmarquez@mdanderson.org (C.M.); csullivan@mdanderson.org (C.R.S.); 3Office of Health Policy, The University of Texas MD Anderson Cancer Center, 7007 Bertner Avenue, Unit 1677, Houston, TX 77030, USA; dmbenson@mdanderson.org

**Keywords:** mobile mammography, mammography, radiology, breast cancer, public health, screenings

## Abstract

Breast cancer, the second most common cause of cancer in women, affects people across different ages, ethnicities, and incomes. However, while all women have some risk of breast cancer, studies have found that some populations are more vulnerable to poor breast cancer outcomes. Specifically, women with lower socioeconomic status and of Black and Hispanic ethnicity have been found to have more advanced stages of cancer upon diagnosis. These findings correlate with studies that have found decreased use of screening mammography services in these underserved populations. To alleviate these healthcare disparities, mobile mammography units are well positioned to provide convenient screening services to enable earlier detection of breast cancer. Mobile mammography services have been operating since the 1970s, and, in the current pandemic, they may be extremely helpful. The COVID-19 pandemic has significantly disrupted necessary screening services, and reinstatement and implementation of accessible mobile screenings may help to alleviate the impact of missed screenings. This review discusses the history and benefits of mobile mammography, especially for underserved women.

## 1. Mobile Mammography Units

In the United States, the American College of Radiology and the Society of Breast Imaging recommend screening mammography every year for women 40 years and older. Regular mammographic screenings detect breast cancers at early stages, when interventions are more effective, leading to better survival outcomes [1]. Women who undergo mammography infrequently or not at all are more likely to be diagnosed with more advanced breast cancer, which is associated with poorer clinical outcomes [2]. This highlights the importance of regular screening mammography. 

Mobile mammography units (MMUs) have been used for six decades. In 1960, Dr. Philip Strax operated the first self-contained mammographic unit as part of a controlled trial to assess whether screening mammography could reduce mortality [3]. The study found a decrease in mortality through 18 years of follow-up. In the 1970s, Dr. Edward Sickles helped streamline costs to make mammography more accessible and helped promote the use of mobile vans in the United States [3].

Early implementation of MMUs has demonstrated many initial benefits. While the initial start-up costs were significant, long-term implementation could be cost-effective due to the lack of rent-related expenses [3]. Furthermore, MMUs enabled both working and nonworking women to undergo screenings in an accessible location efficiently, whether in between work hours or “just before grocery shopping” [3].

Mobile units can lessen the impact of disparities by reaching women who may be unable to travel to in-person clinics, and MMUs have been shown to be effective for particular subgroups of women, such as those older than 60 years [4]. Types of MMUs include vans (Figure 1 and Figure 2), recreational vehicles, and traveling clinics; thus, MMUs can provide necessary screening services in both urban and rural environments [5]. MMUs have also been able to generate significant participation among women, and they were effective in detecting abnormalities on screening examinations [6]. Another report showed that mobile mammography identified numerous low-grade invasive estrogen receptor- and progesterone receptor-positive tumors that were treatable with early interventions [7]. 

## 2. Breast Cancer and Mammography in Underserved Women

By providing access to the detection of early-stage tumors, mobile mammography services can help to prevent advanced-stage malignancies in vulnerable populations, defined as those who receive fewer healthcare services, encounter economic, cultural, or linguistic barriers, are unfamiliar with healthcare delivery service, or face a shortage of providers [8]. Underserved and vulnerable populations are more heavily impacted by breast cancer compared with nonvulnerable populations.

Other social determinants, such as socioeconomic status, culture, and social injustice, can also impact access to healthcare services [9]. While studies have shown that early cancer prevention, detection, and treatment are needed for improved outcomes, vulnerable populations may have barriers to accessing these services [10]. For example, women in vulnerable populations have less access to education on breast health, are less likely to access screening services necessary for early detection, are more likely to experience delays in treatment or have incomplete treatment (such as missed adjuvant chemotherapy), and have poorer long-term survival outcomes. Women of lower socioeconomic status or minority backgrounds have a greater risk of being diagnosed with advanced-stage breast cancer and lower rates of survival [10,11,12,13,14,15,16]. Currently, mortality rates of Black women diagnosed with breast cancer are 40% higher than those of White women [17].

Underserved populations have various barriers to obtaining breast cancer care throughout the care continuum. For example, lower access to preventive care services, less participation in trials, inadequate follow-up, communication barriers with providers, and racism may impact the ability of underserved populations to receive the optimal services associated with improved breast cancer outcomes [12,14,18].

While it may be said that breast cancer does not discriminate between the rich and the poor, studies have found otherwise. Women living in poverty are more likely to be diagnosed with breast cancer in advanced stages [2]. Other studies have found that women living in poverty were 1.46 times more likely than women not living in poverty to suffer death as a sequela of their breast cancers [18]. These statistics illustrate that, while breast cancer may affect anyone, women with lower socioeconomic status are particularly vulnerable to poorer outcomes [19].

Women identifying as American Indian, Hispanic, and Black have been shown to have decreased use of mammography services [20,21]. Studies have found that screening rates were 19% lower in Black women than in White women [18]. Studies have also shown that women 55 years and older have been less consistent in obtaining screening mammograms [1]. MMUs can address these disparities. The convenience of mobile services in providing screenings closer to home or work has been shown to appeal to Black women [20]. A French study found that individuals older than 70 years in more remote and underserved regions preferred mobile mammograms to screenings in radiology offices [22]. Another study in the Appalachian region in the United States found that a mobile unit was able to reach women who lived in rural areas and had a history of missed screenings, suggesting that a vulnerable population can be supported through such programs [23]. One study found that only 26% of low-income underinsured women utilized free available screening programs [2]. This suggests that other factors are limiting the use of screening programs besides costs, such as logistics and convenience. These studies illustrate that MMUs can reach women who might miss regular screenings.

Mobile mammography services may advertise through outreach efforts at community events and fairs, but this may not always be sufficient. One study found that reliance on patient self-referral for services was inadequate in addressing communities with a higher risk of underutilization of mammography services [24].

## 3. Patient Factors Affecting Selection of Mobile Mammography

To better understand the utilization of mobile mammography services, studies have assessed the demographics and outcomes of women who select MMUs. An American study found that the patients undergoing mammography at a cancer center were significantly different than the patients who underwent mammography in a mobile unit [25]. More White patients elected to be screened at the cancer center, while more Black and Hispanic patients chose to be screened on the mobile units [25]. Patients screened at the cancer center were more likely to adhere to screening mammography guidelines, and patients using the mobile units were less likely to return for follow-up imaging studies [25].

Analyzing the demographics of women who elect mobile services can also indicate future directions and opportunities. Mobile mammography programs have imaged women with high rates of obesity and with comorbidities, which allows for early intervention for other preventive services besides mammography [26]. Demographics also differ regarding women who are likely to return to mobile units. A case–control series found that uninsured Black women ages 50–65 years were more likely to have a return visit to MMUs, while women with rural zip codes or who were unemployed were less likely to have a return visit to MMUs [27]. Another study found that Black women were more likely to have significant use of MMUs [28].

## 4. Operational Considerations for Mobile Mammography Programs

There are multiple organizational and funding models regarding the implementation of mobile mammography services, and the approach used depends on local variables and demographics. Payment systems include grant funding, philanthropy, and fee-for-service arrangements. Some MMUs accept self-referrals while others do not. Self-referred patients may need a provider to be responsible for the patient’s clinical breast exam. Many MMUs require a prior clinical exam, such that patients with negative clinical breast exams proceed to screening mammography, while women with positive clinical exams undergo diagnostic mammography.

Although mobile mammography programs have high start-up costs, which may impede their implementation [29], studies have found that optimizing mobile mammography infrastructure and workflow can help to offset initial limitations [30]. A study by Carkaci et al. detailed some of the optimization strategies used at the University of Texas MD Anderson Cancer Center (MDACC). The MDACC Mobile Mammography Program ensures adequate patient participation with preregistration through phone appointments instead of a walk-in system [30]. Furthermore, abnormal screenings are closely tracked until appropriate follow-up is achieved [30].

An important component of mobile mammography is appropriate follow-up. Use of mobile mammography services has been associated with poor follow-up rates [18]. Mobile users were less likely to return to the same unit where they were screened [18]. One of the known barriers to follow-up is accessibility, which can be impacted heavily by communication barriers. Underserved women may lack access to phones and email or have no permanent home address, preventing adequate communication [7].

Requiring preregistration may be seen as a barrier to obtaining mobile mammography services. On the other hand, preregistration allows for demographic and reimbursement information to be inputted prior to the mammography appointment. In addition, patient preregistration helps guarantee that the mobile mammography provider will have at least a suitable minimum number of patients registered prior to committing resources to provide mammography services at a site.

The MDACC Mobile Mammography Program participates in Project VALET (Valuable Area Life-Saving Exams in Town), which targets underserved women in the greater Houston metropolitan area. Project VALET patients are preregistered, and mobile screening mammography is coupled with the dissemination of educational materials on breast health [7]. In the MDACC Mobile Mammography program, patient registration occurs through scheduling with either corporate sites that provide insurance-covered mammograms or through clinics that are in direct communication with patient service providers, who can schedule Project VALET patients. These patient service providers ensure that the schedules and templates are set up efficiently. In Project VALET, each patient appointment is scheduled for 15 min, with 8 min dedicated to the patient encounter and 7 min of imaging time [7].

With Project VALET, patients are seen from 7:30 a.m. to 3:30 p.m., and technologists start with a pre-inspection of the van before driving to the site. Thereafter, quality control routines are performed at each site. Typically, two technologists are involved at each site.

Although mobile mammography services have numerous benefits, there are significant limitations regarding their implementation. Major limitations include equipment breakdowns and inclement weather. The logistics of developing and managing operations may pose challenges. It is thought to be helpful if patient preregistration, internet connectivity, patient education, and follow-up appointments are ensured prior to the mobile mammography appointment [31].

In the MDACC Mobile Mammography Program, approximately one-third of cases are imaged with digital mammography, and approximately two-thirds of cases are imaged with digital breast tomosynthesis (DBT). The current plan is to convert all imaging to DBT, as DBT has been shown to be associated with a decrease in the recall rate and an increase in the cancer detection rate [32]. Following image acquisition, the images are sent through a secure wireless connection to the hospital, where the images are interpreted the next day by fellowship-trained breast imagers with the use of Image Checker^®^ 2D CAD Technology (Hologic, Inc., Marlborough, MA, USA), a computer-aided detection system. Computer-aided detection of breast lesions involves the use of computer schemes to mark suspicious findings on mammography, and the interpreting radiologist then determines if further evaluation is needed [33].

## 5. Mammography in the COVID-19 Era

The relevance of mobile mammography services has been heightened during the COVID-19 pandemic. The current pandemic has had a disruptive effect on preventive health. To reduce exposure to staff and patients, lower-priority examinations, which include breast cancer screenings, were delayed at many medical centers [34]. During the initial pandemic wave from March 2020 to September 2020, there were significant delays in breast cancer screenings and delays in breast cancer diagnoses [35]. Mobile mammography screenings have been suggested as a way to improve screening mammography participation [36]. Individuals less likely to return for screenings during the pandemic were younger, uninsured women living in underserved regions with barriers such as travel distances and the need for an interpreter [37]. One study found that women without insurance, women of Black or Hispanic ethnicity, and women aged over 53 years were more likely to cancel mammography appointments during the pandemic [38]. Similarly, fewer cancers were detected in Asian, Hispanic, and Black women than White women during the pandemic, correlating with pandemic-related delays in detection [35]. The significance of these interruptions in health maintenance may severely impact a woman’s health, and identifying more accessible screening services is crucial.

Although the initial costs associated with developing a mobile mammography program may seem to be high, MMUs are accessible and effective in helping to identify earlier-stage breast cancers and in mitigating some healthcare disparities [39,40,41].

## 6. Global State of Mobile Mammography Screening

The global impact of breast cancer is significant as it represents the leading cause of cancer death in women worldwide [42]. Low–middle-income countries (LMICs) have a higher cancer burden and higher mortality from breast cancer compared with high-income countries (HICs). The 5 year survival rate for breast cancer is 90% in HICs, but it is only 10–40% in LMICs [42]. In HICs, 75% of breast cancers are diagnosed at earlier stages, while, in LMICs, 75% of breast cancers are diagnosed at later stages (stage III or IV) [42,43].

While screening mammography effectively reduces breast cancer mortality by facilitating early diagnosis, there are different approaches to screening mammography in each country. Some countries have developed breast cancer screening guidelines according to the needs and resources specific to their population. The Breast Health Global Initiative (BHGI) was established in 2002 as an international health alliance of 40 countries and tasked with developing evidence-based, economically feasible, and culturally sensitive guidelines for LMICs to improve breast health outcomes [44]. Many LMICs without country-specific breast cancer screening guidelines adopt the recommendations of the BHGI. The BHGI advocates a stratified approach to breast cancer screening guidelines based on classifying the available national resources into four levels: basic to maximal. For regions with only basic or limited resources, screening is not prioritized, and the goal is early detection using clinical breast examinations and prompt diagnostic evaluation of symptomatic patients. For regions with third-tier resources available, mammographic screening may occur every 2 years in women aged 50–69 years and possibly annually in women aged 40–49 years. For regions with maximal resources available, annual mammographic screening may occur in women 40 years and older. The BHGI suggests that women 50–69 years old should have mammographic screening every 2 years, even in countries with limited resources [44].

Many HICs have organized national breast cancer screening programs where all eligible women are invited to participate in screening. Some countries participate in opportunistic screening in which women participate in screening by self or clinician referral, as able, given availability and cost.

There is an intrinsic disparity in the mammography resources between LMICs and HICs, and some LMICs may not be able to afford mammography equipment, maintenance, and workforce training [42,45]. LMICs may not have the infrastructure to support national screening guidelines, to support the healthcare teams to provide screening services, or to provide necessary community education [42]. Patients in LMICs may also lack the resources to attend screening services because of transportation and examination costs or other financial constraints [46]. Cultural barriers such as stigma, fatalism, inadequate knowledge about the importance of screenings, or distrust of healthcare providers may discourage screenings [39,43,44,45,46,47].

To address barriers to screening, countries of various economic levels have implemented mobile mammography with mixed results. Successful mobile mammography programs have been implemented in LMICs such as Jordan, Egypt, and Brazil [41,47,48,49,50,51,52]. Examples of national screening initiatives are described below.

**France:** A national screening program invites every woman 50–74 years old to obtain a mammogram every 2 years. Women are provided a list of radiology offices from whom they may select a provider. Despite mammography being free to women, there continue to be social, cultural, systemic, and behavioral barriers to participation, with low socioeconomic status and rurality correlating with low participation [24]. In one study, women in a region covered by MMUs were provided the option to obtain mammography at a radiologist’s office or through MMUs. Women invited to mobile mammography screening had increased participation (60%) compared with screening in radiologists’ offices (42%); this was more pronounced in individuals older than 70 years and in more remote or underserved regions. Women farther from radiology offices had lower participation in screening. The study’s authors suggested that geographic inequalities may be improved by the addition of MMUs.

**Brazil:** The adoption of screening guidelines varies. For some organizations, the guidelines recommend mammography every 2 years for women 50–69 years old and annual clinical breast examination (CBE) in women 40–49 years old [53,54]. Other guidelines suggest mammography for all women 40 years or older. Funding for mammography may be covered by the government or, for women with higher socioeconomic status, private insurance [55]. While women may self-refer for mammography, they are typically educated on the need for screening by primary care physicians who encourage screening. There are documented disparities in screening utilization in rural versus urban areas. One program specifically introduced mobile mammography to rural areas [48,55]. Beginning in 2003, a government-organized and -funded screening program was created using mobile and fixed mammography units in the Barretos region of Sao Paulo, Brazil. This region is made up of 19 cities with approximately 54,000 eligible women. The program had access to one mobile unit and three mammography machines in a fixed unit. MMUs could also provide cervical cancer screening. Women 40 to 69 years old were invited to participate in screening at MMUs or fixed units. Thirty-one percent of all women eligible for screening participated in the program, with the mobile unit accounting for 59% of the examinations performed. MMUs conducted a daily average of 26 examinations versus 15 daily examinations for the fixed unit. Mobile units were shown to be critically important to overcoming barriers in remote areas with a lack of resources and difficulty in accessing the public health system. Patients indicated that home visits by local community agents were the most effective strategy for encouraging participation [55].

Following the success of this study, screening has been expanded to 108 municipalities (targeted population of 223,000 women), and more mobile units have been added. In the follow-up study, MMUs served an average of 60 patients a day, and 54% of eligible women were screened by mobile mammography [54]. Barriers included a shortage of primary physicians to provide breast awareness and referrals for mammograms [55].

**Nigeria:** Currently, Nigeria has no national breast cancer screening guidelines. On an individual and a facility basis, there may be piecemeal adoption of international screening practices based on guidelines from the United States or Europe [39]. Some facilities or nonprofit organizations provide mobile screening with CBE. Most healthcare costs are borne by the individual, with few private health insurance companies available. There is a National Health Insurance Scheme which covers less than 5% of the population. Opportunistic mammographic screening is available with low uptake (estimated at 3%). Given the low resources in the country, community screening typically involves CBE in outreach programs, the uptake of which is also poor. A survey was performed on women over 40 years old in two Nigerian cities, one with mammography services readily available and the other without direct access to mammography (but available in the former community 90 min away). Similar breast cancer awareness was documented in both groups. Only 11% of women in either city were aware of mammography. Breast cancer screening had been recommended to 37% of women; however, only 2–3% had undergone mammography. Lack of awareness and lack of perceived need were cited as the most common reasons for not undergoing recommended screening. Cost, fatalism, and lack of access were also identified as barriers. In the two groups, only 20% and 27% had undergone CBE in the past, despite its availability in both communities. CBE was identified as increasing the likelihood of mammography use. This survey study illustrated that another barrier against adequate utilization of resources is the community perception of the necessity of screening [39]. Community education on the need for mammographic screening is crucial to improving uptake of screening mammography services.

**India:** As recently as 2016, the Indian government created a population-based cancer screening program for woman older than 30 years for CBE with referral to health centers for mammography if needed [56]. Prior to this, beginning in 2010, the international radiology nonprofit group RAD-AID partnered with Phillips Healthcare and a government medical center in Chandigarh, India, to create a mobile women’s imaging program for semiurban and rural India [40]. Within this program, women could access women’s health education, screening mammography, osteoporosis screening with dual-energy X-ray absorptiometry, and virtual colposcopy. Local government policies provide free medical care for women of the lowest social economic means. With strong local media support, patient turnout has been high, with 20,000 patients seen between 2012 and 2017. Patients with abnormal mammography findings were found to have an 83% follow-up rate [57,58]. Important components of developing robust screening mammography programs are training technologists at frequent, short-term intervals and implementing a system to assess overall quality improvement. Within this screening program, a technologist training study was performed. A technologist training course was introduced, and radiologists using objective quality criteria scored mammographic studies before and after the intervention. There was quality improvement in the short term; however, after 6 months, scores returned to baseline pretraining levels [40]. Continuing education is now provided on a recurring virtual basis for radiologists and technologists through RAD-AID. In addition, the program partners with a local nursing school for staffing needs. Barriers to screening uptake in India include infrastructure, fatalism, religion, caste, and education [56].

**United States:** There are varying breast cancer screening guidelines. The American College of Radiology and the Society of Breast Imaging recommend annual mammographic screening beginning at age 40 years for women at average risk [59]. The American Cancer Society (ACS) states that women between 40 and 44 years have the option to undergo screening mammography every year. The ACS recommends that women aged 45 to 54 years should get mammograms every year. The ACS states that women aged 55 years and older can switch to mammograms every other year or they can continue with annual mammography [59]. There are various payment patterns for healthcare, predominantly through private insurance, but older women and women of low socioeconomic status may qualify for government insurance. Some women of low socioeconomic status are uninsured or underinsured.

Women identifying as American Indian, Hispanic, or Black have been shown to have decreased use of mammography services [22,23]. Studies have also shown that older women have been less consistent in obtaining screening mammograms [21].

Mobile mammography services are available through for-profit and nonprofit organizations. For-profit mobile mammography services may be employed to increase access for professional women who are insured. Nonprofit mobile mammography services may be funded by government programs or grants to provide mobile mammography screening to underserved populations.

The convenience of mobile services in providing screening closer to home or work has been shown to appeal to Black women [22]. Another study found that a mobile unit was able to reach women in rural areas with a history of missed screenings, suggesting that a vulnerable population can be supported through these programs [25]. A large public safety net system (providing care to uninsured and underinsured) instituted mobile mammography screening [26]. Community sites such as health centers, organizations, public housing, health fairs, and private companies could request a visit by the MMU. Women were self-referred, not required to list a primary care physician, would receive results by mail in 14 days, and, if needed, could follow up at diagnostic facilities in the network. The study found that this program was predominantly used by women of ethnic minorities; only 12% of patients were White. Only 1.6% of the mobile mammography visits were requested by private businesses, with the highest mobile mammography use by community health centers (51%). Fifty-six percent of women with abnormal screening results did not have diagnostic follow-up or had unknown follow-up status. Forty percent of patients chose follow-up care outside the follow-up system (as they had private insurance). This suggests that MMUs may be attractive to women outside of the underserved population. Self-referral by women meant that some of the limited public health resources were being used by women with means. One study found that reliance on patient self-referral for services was inadequate in addressing communities with a higher risk of underutilization for mammography services [26]. Alternative mobile mammography models, such as Project VALET, utilize a funded preregistration system that can offer services for communities that may have a higher risk of underutilization [7]. With increased funding, MMUs can have an increased capacity to serve vulnerable communities.

## 7. Discussion

Screening mammography has been shown to be associated with decreased mortality from breast cancer. The efficacy of screening mammography has been hindered globally by disparities in access to mammography services. The COVID-19 pandemic has further exacerbated healthcare disparities in screening mammography. MMUs can help alleviate these disparities by providing convenient screenings. Since the 1970s, MMU have been shown to be beneficial by identifying early stage, nonpalpable breast cancers. MMUs can be a valuable resource in identifying early breast cancer and in reducing healthcare disparities.

The implementation of MMUs for screening requires significant organization and management of logistics, and often involves major start-up costs. Careful, detailed planning and effort is required to ensure that MMUs can run successfully. Mission trips that aim to help alleviate access to screening mammography in developing countries must also make strides to collaborate on sustainable educational programs, including training local technologists and radiologists. Such methods have been found to be successful, as noted by RAD-AID trips and studies in global communities.

Disparities in access to mammography extend internationally. Limited access to screening mammography is common in many LMICs. MMUs may be beneficial for breast cancer screening in LMICs as they can go to under-resourced rural areas and outlying towns [40,48]. In addition, they provide a streamlined experience (for women with limited time away from other responsibilities) and eliminate some logistical issues associated with navigating an unfamiliar healthcare system [47]. If organized in concert with local providers, mobile mammography can be coupled with educational experiences, translation services, and culturally sensitive practices [40]. Due to limited options in remote areas, an annual screening mammography program could provide consistency, as many women indicate preference for mobile mammography and are unlikely to have screenings at other locations [24,49].

Best practices for developing, implementing, and improving mobile mammography screening services worldwide may be beneficial for regions with persistent disparities in access to mammography. Some strategies that have increased success in mobile mammography programs include funding for examinations and follow-up care, coupling mammography with educational efforts, partnership with trusted local community leaders to encourage participation, proper selection of the target population (with rural and older populations demonstrating higher participation), packaging of additional screening services such as cervical cancer and osteoporosis screening with mammography, and preregistration rather than walk-in access.

Since breast cancer awareness and education are important goals in every country, the ability to provide education to under-resourced communities could help meet goals for downstaging breast cancer diagnosis [42,47]. Development of appropriate guidelines could encourage mobile mammography efforts in developing nations with existing support infrastructure [47,49,50,51].

## Figures and Tables

**Figure 1 diagnostics-12-00902-f001:**
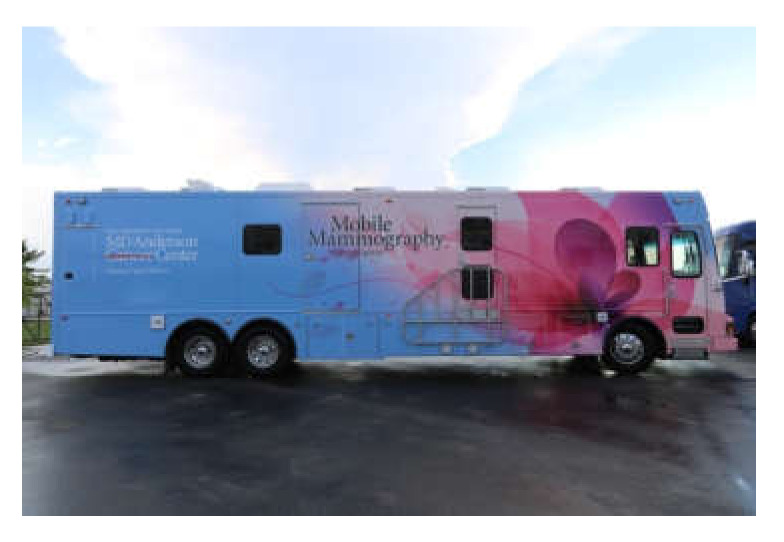
A mobile mammography van.

**Figure 2 diagnostics-12-00902-f002:**
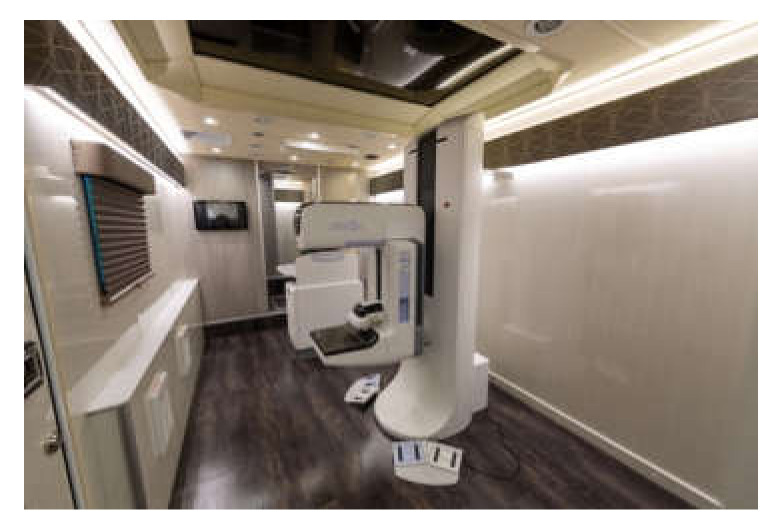
Interior of a mobile mammography van, with a mammography unit seen centrally.

## Data Availability

Not applicable.

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
