# Peer review of "Mobile Mammography Services and Underserved Women"

_diagnostics, 2022, doi:10.3390/diagnostics12040902_

Round 1
Reviewer 1 Report
Line 30: The authors state: The 30 study found a decrease in mortality through 18 years of follow-up. Can they please expand on what the reduction was and whether this just relates to mobile mammography (was this via and RCT) or mammography in general (eg HIP RCT).
In the introduction, the authors discuss other barriers to mammography among women. Socioeconomic status is mentioned as one of these barriers however there is no tying in of this two mobile mammography. Even if mobile mammography units are readily available increasing convenience what are the cost implications for the individual who have low social economic status?
It might also be useful given this is going to be published in an International Journal to give some background on how mammograms are costed in the states. Are they publicly funded or are they provided on a user pays basis? - I later came across this statement on line 93: “One study found that only 26% of low-income 93 underinsured women utilized free available screening programs” so describing how mammograms are provided is really important. Finally came across how provide at the top of page 4. Recommend even just a sentence early in the introduction indicating there is variance in how programs are funded
Line 82 the authors describe current US guidelines for screening i.e. every 12 months. This is quite frequent compared to other settings for example every three years in the UK and every two years in Australia. There could be further discussion therefore around this screening frequency to increase appeal to international audiences
Line 91 the authors state that there is a higher prevalence of breast cancer in Latina and black women. This statement needs to be supported by statistics
Line 122 The authors state: “Mobile mammography programs have been shown to have imaged women with high rates of obesity and with comorbidities, which allows for early intervention for other preventative services besides mammography”. The authors should discuss why these women prefer mobile screening
Throughout I think there could be better use of subheadings (Is 3 missing as it jumps from 2 o 4)
In the section on Screening Mammography in Developing Countries, it is important to give some examples of the screening programmes available in those settings as many of those countries do not have any formalised screening programmes. How mobile mammography may/not fit with this is important to acknowledge is at the moment it seems like a vision rather than a realistic proposition
Should section 5. Discussion be section 6 conclusion?
Author Response
Line 30: The authors state: The 30 study found a decrease in mortality through 18 years of follow-up. Can they please expand on what the reduction was and whether this just relates to mobile mammography (was this via and RCT) or mammography in general (eg HIP RCT).
- This line was clarified to indicate that more frequent screenings as per the screening trial that utilized a mobile unit, was associated with long-term reduction in breast-cancer related death.
In the introduction, the authors discuss other barriers to mammography among women. Socioeconomic status is mentioned as one of these barriers however there is no tying in of this two mobile mammography. Even if mobile mammography units are readily available increasing convenience what are the cost implications for the individual who have low social economic status?
- Cost implications are a crucial aspect of these barriers. Some mobile mammography examinations are funded through insurance and can be geared for professional women. For uninsured women, funded mobile mammography units can provide free screenings. However, one of our reviewed studies show that pricing is not the only barrier involved, and this is where we have found that mobile units can be uniquely positioned to provide not only affordable but also convenient screenings.
It might also be useful given this is going to be published in an International Journal to give some background on how mammograms are costed in the states. Are they publicly funded or are they provided on a user pays basis? - I later came across this statement on line 93: “One study found that only 26% of low-income 93 underinsured women utilized free available screening programs” so describing how mammograms are provided is really important. Finally came across how provide at the top of page 4. Recommend even just a sentence early in the introduction indicating there is variance in how programs are funded
- We discussed funding models in the first two paragraphs of Section 4. Operational Considerations for Mobile Mammography Programs.
Line 82 the authors describe current US guidelines for screening i.e. every 12 months. This is quite frequent compared to other settings for example every three years in the UK and every two years in Australia. There could be further discussion therefore around this screening frequency to increase appeal to international audiences
- We address this discussion by discussing US guidelines compared to the other countries, and we include both LMIC as well as HIC.
Line 91 the authors state that there is a higher prevalence of breast cancer in Latina and black women. This statement needs to be supported by statistics
- References were included for the studies that illustrated this statistic.
Line 122 The authors state: “Mobile mammography programs have been shown to have imaged women with high rates of obesity and with comorbidities, which allows for early intervention for other preventative services besides mammography”. The authors should discuss why these women prefer mobile screening
- We did not indicate that these women were found to indicate a clear preference for mobile services. Instead we explain that this population is screened with mobile mammography and this creates an opportunity for mobile visits to include other preventative services.
Throughout I think there could be better use of subheadings (Is 3 missing as it jumps from 2 o 4)
- This was corrected.
In the section on Screening Mammography in Developing Countries, it is important to give some examples of the screening programmes available in those settings as many of those countries do not have any formalised screening programmes. How mobile mammography may/not fit with this is important to acknowledge is at the moment it seems like a vision rather than a realistic proposition
- We included a few examples in Section 6. State of Mobile Mammography Screening Globally.
Should section 5. Discussion be section 6 conclusion?
- We adjusted the Discussion section. The Discussion is now Section 7.
Reviewer 2 Report
The authors of this paper focus on findings correlating decreased usage of screening mammography services in underserved populations. The paper is easy to read but turns around some repeating items, respectively, MMU (Mobile Mammogram Unit) and socio-economic aspects.
Although the topic suits the journal and its scope, I believe that some more in-depth analysis could strengthen the scientific grounds of the contribution.
- Some tables should be added to summarise the most meaningful statistics out of the scientific literature about MMU and socio-economics aspects across all the countries mentioned in the sections.
If the authors aim to tackle a correlation analysis between underserved populations during the covid-19 pandemic, I strongly suggest adding a more comprehensive section.
- Discussions are barely described. I would expect some more considerations summarising all the aspects tackled throughout the sections in the paper, especially for a review.
Author Response
The authors of this paper focus on findings correlating decreased usage of screening mammography services in underserved populations. The paper is easy to read but turns around some repeating items, respectively, MMU (Mobile Mammogram Unit) and socio-economic aspects.
- The use of the MMU term was initially defined as mobile mammography units in the second paragraph of the paper, such that MMU could be used as a shorter abbreviation for increased clarity within the piece. There are a few points re-emphasized in the piece to highlight the crucial need for breast cancer services.
Although the topic suits the journal and its scope, I believe that some more in-depth analysis could strengthen the scientific grounds of the contribution.
- We added more details to provide more analysis to support, specifically in the applications in the Covid-19 era (Section 5) and international health (Section 6).
Some tables should be added to summarise the most meaningful statistics out of the scientific literature about MMU and socio-economics aspects across all the countries mentioned in the sections.
- We appreciate the suggestion to add a table. However, we decided to expand Section 6 (State of Mobile Mammography Screening Globally) in order to provide more detailed information on mobile mammographic screening in France, Brazil, Nigeria, India and the United States.
If the authors aim to tackle a correlation analysis between underserved populations during the covid-19 pandemic, I strongly suggest adding a more comprehensive section.
- We expanded on the section on the COVID-19 pandemic and its impact on mammography (Section 5).
Discussions are barely described. I would expect some more considerations summarising all the aspects tackled throughout the sections in the paper, especially for a review.
- Our discussion section (Section 7) was expanded.
Round 2
Reviewer 2 Report
I appreciate the authors tackled most of the points highlighted in my previous review. I want to thank them for providing a newer version of their paper, which shows higher quality under several perspectives.
In my opinion, the current version is almost ready for publication in Diagnostics.
My only suggestion is now in regards to section 3 highligthing some aspects related to pre-screening impact on mammogram analysis.
Nowadays, several screening steps for breast cancer rely on CAD (Computer Aided Diagnosis) and CADx (Computer Aided Detection) systems, which deal with MRI and Mammogram analysis using Computer Vision and Artificial Intelligence techniques. Some studies thouroughly tackle these two points. I believe these two articles below would provide the paper with a 360° analysis of the main aspects related to mammograms.
Here are the links to the two studies:
https://www.ncbi.nlm.nih.gov/pmc/articles/PMC7528986/
https://link.springer.com/chapter/10.1007/978-3-319-68560-1_63
The ones above represent only minor issues.
Author Response
We appreciate your suggestions. We have incorporated a section (penultimate paragraph in Section 4) discussing the workflow in the MDACC Mobile Mammography Program, including the use of digital breast tomosynthesis (DBT) and computer-aided detection.